# Manipulate Large Language Models in Time Series Forecasting by Token Disruption

## Abstract

Although Large Language Models (LLMs) have demonstrated substantial potential as powerful zero-shot time series forecasters, recent evidence shows that even small adversarial perturbations can significantly degrade their performance under strict black-box settings. However, existing attacks typically rely on repeated queries to the target LLM forecaster, making them easily detectable and anomalous in real-world scenarios. To overcome this limitation, we introduce the Token Disruption Attack (TDA) that generates perturbations by solely querying the local tokenizer rather than directly interfering with the model. We first formulate the attack as a non-convex optimization problem that maximizes the divergence in encodings produced by the target tokenizer, and then design a dynamic programming–based method to solve it efficiently. By injecting subtle perturbations into the raw time series, TDA induces substantial distortions during tokenization, which subsequently propagate through the model and ultimately result in severe forecasting errors. Extensive experiments on ten LLM-based and two non-LLM-based forecasters across six applications demonstrate that minor perturbations can cause large downstream distortions, leading to forecasting errors that increase by nearly 20%.

## 1 Introduction

Time series forecasting serves as a fundamental analytical tool across diverse fields, including finance, transportation, energy management, and climate science. By leveraging historical data to identify underlying patterns and trends, forecasting models facilitate the prediction of future events, enabling data-driven decision-making (Zhuang et al., 2022). Accurate forecasts are essential for optimizing resource allocation, enhancing operational efficiency, and mitigating potential risks, making them indispensable for strategic planning (Lim & Zohren, 2021; Liu et al., 2022b).

Large Language Models (LLMs) have recently shown remarkable potential for capturing complex temporal dependencies in time series forecasting (Garza & Mergenthaler-Canseco, 2023; Jin et al., 2024). Their ability to integrate contextual information and model long-range correlations makes them particularly well-suited for prediction tasks across diverse domains (Brown, 2020; Touvron et al., 2023). By reformulating time series forecasting as a next-token prediction problem, recent studies demonstrate that LLMs can serve as powerful foundation models for time series, primarily due to their emerging capacity for zero-shot forecasting. The zero-shot ability of LLM-based forecasters enables them to generalize across a wide range of tasks without the need for extensive task-specific retraining (Rasul et al., 2023; Liang et al., 2024; Ansari et al., 2024).

Recent studies have raised concerns about the reliability of LLMs in time series forecasting, demonstrating that these models can be adversarially manipulated using imperceptible perturbations (Liu et al., 2025; Zhang et al., 2025). Unlike adversarial attacks in thoroughly-investigated domains such as computer vision (CV) and natural language processing (NLP), attacking LLM-based forecasters face new challenges due to the absence of ground truth and the strict black-box nature of LLMs. Specifically, to prevent information leakage, ground truth remains inaccessible to both forecasters and attackers. For example, in a 5-minute-ahead Google stock price forecasting task, the actual stock price at 11:05 AM is inherently unknown at 11:00 AM, making it impossible to leverage true labels to generate adversarial perturbations, an approach commonly used in static domains (Goodfellow et al., 2015; Zheng et al., 2016). Furthermore, LLMs often operate as proprietary, restricting attackers from accessing internal parameters, architectures, or training data. This nature calls for adversarial

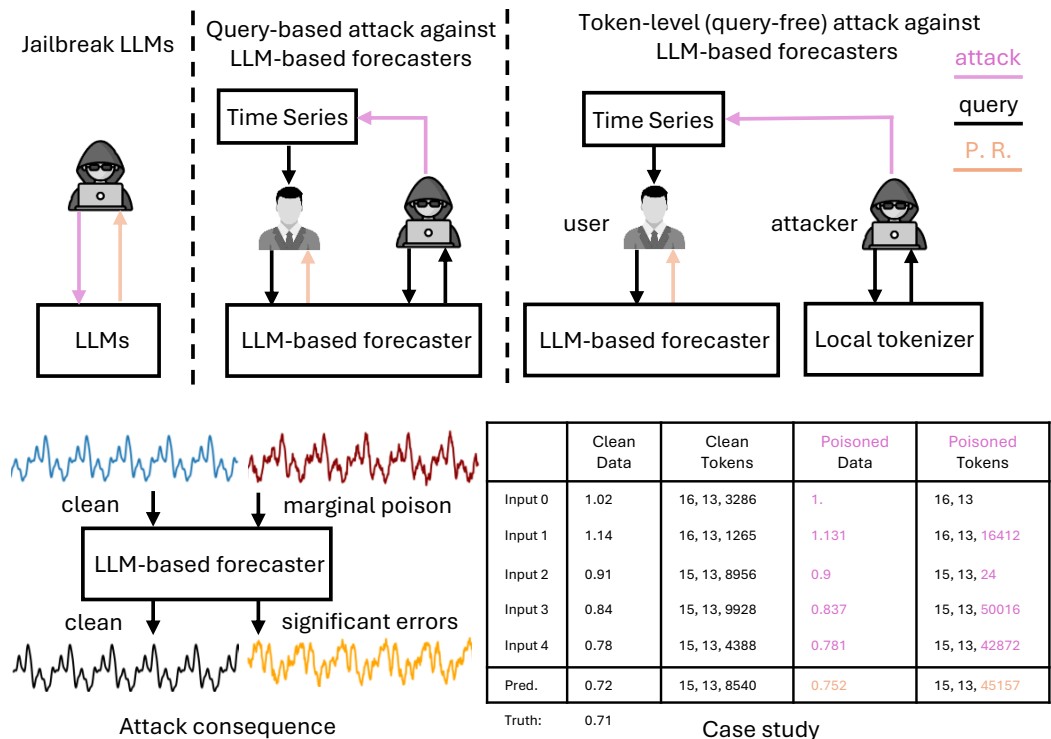

Figure 1: *Top*: Conceptual comparison between three scenarios: (i) jailbreaking LLMs, (ii) query-based attacks against LLM-based forecasters, and (iii) the proposed TDA, which manipulates LLM-based forecasters by querying only the local tokenizer without direct access to the forecasting model. The orange P.R. denotes poisoned responses. *Bottom left*: Illustration of an attack consequence, where a minor input perturbation propagates into substantial forecasting errors, ultimately compromising the user's decision-making. *Bottom right*: A toy numerical case study of TDA on GPT-4o-based forecaster. With only a 0.87% average input modification, outputs shifted by 3.3% and forecasting errors increased by about 4×. Public examples are provided for both clean and poisoned cases (anonymous). Further details are given in Appendix A.

strategies designed to operate under strict black-box constraints. To address these challenges, studies (Liu et al., 2025; Zhang et al., 2025) have introduced gradient-free, label-free attacks, successfully compromising LLM-based time series forecasters by iteratively querying the target models.

However, a significant **gap** remains in understanding the vulnerabilities of LLMs in time series forecasting, as existing adversarial attacks depend on **iterative queries to the target forecasting model** to approximate gradients and generate adversarial perturbations. This reliance on repeated querying not only introduces computational delays but also compromises the stealthiness rule, which limits the practicality of such attacks in real-time forecasting scenarios. For instance, many LLM-based forecasting models impose access restrictions, preventing attackers from interacting with them in the same manner as legitimate users; the attacker may experience unpredictable delays due to API load or internet connection instability, which can lead to attack failures and undermine the feasibility of real-time exploitation. Given these constraints, a critical question emerges in the exploration of LLM vulnerabilities in time series forecasting: **can an attacker manipulate an LLM-based forecaster without querying the target model?**

We propose a Token Disruption Attack (TDA) to address this gap. LLM-based time series forecasters operate through the inherent next-token prediction mechanism of LLMs, while their associated tokenizers are publicly available and can be easily executed locally (Ali et al., 2024). Rather than directly perturbing the predictions of LLM-based forecasters, TDA reformulates the attack to target the tokenization process, thereby inducing misrepresentations that propagate through the model and ultimately degrade forecasting accuracy. A conceptual comparison between traditional query-based attacks and the proposed query-free TDA is illustrated in Figure 1.

We formulate the tokenization attack as a non-convex optimization problem aimed at maximizing the divergence between token encodings produced by the target tokenizer. To measure the impact of tokenization manipulation, we design a novel metric that incorporates temporal patterns, decimal structures, and delimiters, allowing us to quantify differences after converting time series into tokens. This metric is then embedded into the optimization framework. To efficiently solve the problem, we develop a dynamic programming (DP)–based algorithm that generates adversarial perturbations with minimal computational overhead. TDA eliminates the need to query the target LLM-based forecaster, making it more practical and stealthy under real-world constraints. A toy case study (Figure 1, bottom right) demonstrates the effect of TDA: (i) the average difference between clean and poisoned inputs is only 0.0086, reflecting a highly subtle perturbation; (ii) the prediction shifts from 0.72 to 0.752, a change of 0.032 (representing a $3.7\times$ amplification relative to the input perturbation); and (iii) the original forecasting error of 0.01 increases to 0.042 after the attack, corresponding to a $4.2\times$ amplification in error.

We conduct extensive experiments on state-of-the-art LLM-based forecasting models, including LLMTime (GPT-3.5, GPT-4, Llama 2, and Mistral), vanilla and prompt-based LLM forecasters (GPT-4o, GPT-4o-mini, DeepSeek-R1-Distill-Llama-70B, Llama 70B-Instruct, Gemini 2.5 Flash, and Sonnet 4), as well as additional non-LLM baselines, across six diverse real-world applications. The results show that perturbing as little as 2% of the input data can induce forecasting error increases ranging from 7% to 41%. These findings highlight a critical vulnerability in LLM-based time series forecasting: even minor perturbations to the raw inputs can cause significant distortions during tokenization, which then propagate through the model and lead to substantial forecasting errors.

## 2 RELATED WORKS

**Adversarial attacks against LLMs** have gained increasing attention in text-based tasks, where subtle input manipulations can drastically alter model outputs. Key categories include jailbreak prompting (Wei et al., 2024; Xue et al., 2024), gradient-based attacks (Jia et al., 2024), and embedding perturbations (Schwinn et al., 2024). Recent studies have further highlighted vulnerabilities in the tokenization stage (Fort, 2025), such as watermarking via green tokens (Kirchenbauer et al., 2023), and token-level manipulations in multimodal LLMs (Wang et al., 2024; Yuan et al., 2025), revealing new avenues for adversarial research.

**Adversarial attacks against time series forecasting** pose unique challenges due to the absence of future ground truth (Dang-Nhu et al., 2020), which limits direct optimization. To overcome this, surrogate labels have been used to approximate gradients (Liu et al., 2022a; Lin et al., 2024). However, most prior work assumes white-box access (Xu et al., 2021; Liu et al., 2023; Zhu et al., 2023), making them less practical for real-world black-box forecasters such as LLMs.

**Adversarial attacks against time series classification** differ fundamentally from those on forecasting, and methods designed for one are not directly transferable to the other. The key distinction arises from the autoregressive nature of forecasting, where the ground truth is unavailable at runtime. Consequently, forecasting attacks cannot rely on true labels during adversarial example generation, as this would cause future information leakage. In contrast, classification attacks operate on static, labeled datasets (Karim et al., 2020; Ding et al., 2023; Jiang et al., 2023).

**Adversarial attacks against LLM-based time series forecasting** are typically constrained to black-box settings (Liu et al., 2025; Zhang et al., 2025). Existing approaches often rely on iterative queries to the target model, which leads to computational delays, compromises stealthiness, and raises detection risks. This limitation motivates the development of query-free strategies, an area our work advances through token-level attacks.

## 3 THREAT MODEL

### 3.1 LLMs-BASED TIME SERIES FORECASTING

Consider a $d$-dimensional time series at time $t$, represented as $\mathbf{x}_t \in \mathbb{R}^d$. The historical input sequence is defined as $\mathbf{X}_t \in \mathbb{R}^{d\times T} = \{\mathbf{x}_{t-T+1}, \ldots, \mathbf{x}_t\}$, which consists of the most recent $T$ observations. The true future values for the next $L$ time steps are denoted as $\mathbf{Y}_t \in \mathbb{R}^{d\times L} = \{\mathbf{y}_{t+1}, \ldots, \mathbf{y}_{t+L}\}$.

Let $f(\cdot)$ denote a standard LLM-based time series forecasting model, which generally consists of a tokenization module that transforms raw time series data into a sequence of tokens suitable for processing by the LLM, and a pre-trained LLM that models temporal dependencies and generates predictions autoregressively based on learned representations. The forecasting model $f(\cdot)$ aims to predict future values based on past observations, formulated as:

$$\hat{\mathbf{Y}}_t = f(\mathbf{X}_t), \tag{1}$$

where $\hat{\mathbf{Y}}_t$ represents the model's predicted outputs for the next $L$ time steps. To ensure a balance between predictive accuracy and computational efficiency, the forecast horizon $L$ is typically constrained such that $L \leq T$.

## 3.2 MANIPULATING LLMS IN TIME SERIES FORECASTING

The **objective** of adversarially attacking an LLM-based time series forecasting model $f(\cdot)$ is to induce abnormal outputs that deviate significantly from both the model's standard predictions and the actual ground truth, using minimal and imperceptible perturbations.

We define the adversarial attack as a maximization problem, where the goal is to introduce perturbations that maximize the forecasting error:

$$\max_{\boldsymbol{\rho}} \mathcal{L}\left(f\left(\mathbf{X}_t + \boldsymbol{\rho}\right), \mathbf{Y}_t\right)$$
$$\text{s.t.} \quad \|\rho_i\|_p \leq \epsilon, \quad i \in [t - T + 1, t], \tag{2}$$

where $\boldsymbol{\rho} = \{\rho_{t-T+1}, \ldots, \rho_t\}$ represents the perturbations applied to the clean historical time series $\mathbf{X}_t$, $\mathcal{L}(\cdot, \cdot)$ is the loss function that measures the discrepancy between the model's perturbed predictions and the ground truth, $\mathbf{Y}_t = \{\mathbf{y}_t, \ldots, \mathbf{y}_{t+L}\}$, and $\epsilon$ imposes an $\ell_p$-norm constraint on the perturbations, ensuring that the adversarial modifications remain subtle and difficult to detect.

In real-world forecasting scenarios, the future ground truth $\mathbf{Y}_t$ is inherently unknown at runtime. For instance, in a 5-minute-ahead Google stock price forecasting task, the actual stock value at 11:05 AM is unavailable to both the forecaster and the attacker at 11:00 AM. To prevent future information leakage, the adversary substitutes $\mathbf{Y}_t$ in Equation 2 with the model's predicted values $\hat{\mathbf{Y}}_t$, ensuring that the attack operates without access to future observations.

LLM-based forecasting models typically function as black-box systems, restricting direct access to internal parameters, gradients, and architectures. Consequently, the adversary must operate under strict black-box conditions, where no detailed internal information about $f(\cdot)$ is available. This constraint significantly limits the feasibility of conventional gradient-based attack methods (Alfeld et al., 2016; Kurakin et al., 2018; Karim et al., 2020), necessitating gradient-free strategies for effective adversarial manipulation.

## 3.3 TOKEN-LEVEL ATTACK AGAINST LLMS IN TIME SERIES FORECASTING

In addition to the unavailability of future ground truth and the black-box constraint, we introduce a further restriction: **the attacker cannot query the target model**. This no-query limitation renders gradient-free techniques (Uesato et al., 2018; Ilyas et al., 2019) ineffective, as these typically rely on estimating gradients through repeated querying of the model. To generate effective perturbations in a completely query-free setting, we propose a token-level attack against LLMs in time series forecasting, referred to as the Token Disruption Attack (TDA). Instead of directly manipulating the model's predictions, the attacker targets the tokenization process, introducing subtle modifications that lead to misspecification in tokenized representations. These errors then propagate through the LLM, ultimately distorting the model's predictions. This approach is feasible since most LLMs' tokenization process is open-source and can be executed offline, allowing attackers to analyze and exploit their behavior without requiring direct access to the cloud-deployed target model.

Building on the previous discussion, the **attacker's capabilities and constraints** in this setting can be summarized as follows: **i.** cannot access the training data, the internal structure, or parameters of the LLM-based forecasting model, or the ground truth values. **ii.** cannot query the target model. **iii.** have access to the local tokenizer.

## 4 METHOD

### 4.1 OPTIMIZATION FORMULATION FOR TDA

The proposed method replaces cloud-based queries to LLMs with local tokenizer queries. The **goal** of TDA is to maximize the discrepancy between the tokenized representations of the clean and poisoned time series while ensuring minimal temporal deviation.

Let $\mathcal{I}(\cdot)$ denote the tokenization process that converts a raw time series into a sequence of tokens. Reformulating the attack formulation defined in Equation 2, we express the token-level attack as:

$$\max_{\boldsymbol{\rho}} \mathcal{L}_{\text{tokens}}\left(\mathcal{I}\left(\mathbf{X}_t + \boldsymbol{\rho}\right), \mathcal{I}\left(\mathbf{X}_t\right)\right)$$

$$\text{s.t.} \quad \|\rho_i\|_p \leq \epsilon, \quad i \in [t - T + 1, t], \tag{3}$$

where $\mathcal{L}_{\text{tokens}}(\cdot, \cdot)$ is a loss function that quantifies the discrepancy between the tokenized representations of the clean and perturbed time series. The constraint ensures that the perturbations remain within a predefined norm bound $\epsilon$, maintaining imperceptibility while maximizing the impact on the tokenization process.

### 4.2 DIVERGENCE MEASUREMENT FOR TOKEN SEQUENCES

Directly computing the Kullback-Leibler (KL) divergence or Mean Squared Error (MSE) between clean and poisoned token sequences neglects both the temporal dynamics of time series and the structural composition of tokens, often leading to misalignment. To address this, we introduce a two-level alignment strategy that incorporates explicit temporal and structural position information into tokens. As illustrated in Figure 2, the first level ensures temporal alignment, pairing token sets from the same time step, while the second level ensures structural alignment, matching whole and fractional tokens within each set. This structured alignment prevents incorrect pairwise comparisons and enables more faithful measurement of differences between clean and poisoned sequences.

Figure 2: Temporal and structural alignment for divergence measurement. The number of tokens before or after the decimal point may differ; alignment at the delimiter and decimal point levels ensures structural consistency.

The updated metric for quantifying differences between token sequences is defined as:

$$\text{diff} = \sum_{t=0}^{T-1} \frac{1}{K} \left( \sum_{i=1}^{\tau} \left\| \text{token}_i^{(t)} - \text{token}_i^{*(t)} \right\| + \sum_{j=1}^{K-\tau} \left\| \text{token}_j^{(t)} - \text{token}_j^{*(t)} \right\| \right), \tag{4}$$

where $t$ denotes the time step, $i$ indexes whole tokens, and $j$ indexes fractional tokens. Here, $\tau$ is the number of whole tokens per time step, $K$ is the total number of tokens, and $T$ is the overall sequence length. In this formulation, $\text{token}^{(t)}$ refers to a clean token, while $\text{token}^{*(t)}$ denotes its poisoned counterpart. Vacant positions arising from variable token sequence lengths are padded with zeros.

This metric is inherently robust to variable token lengths in time series inputs, as explicit alignment at the delimiter and decimal point levels ensures structural consistency between clean and poisoned sequences. It is further integrated into the optimization framework described in Equation 3 to compute the attack loss. Since the tokenization process itself is non-convex, the resulting optimization problem also remains non-convex, which poses additional challenges for identifying effective adversarial perturbations.

### 4.3 PERTURBATION GENERATION BY DYNAMIC PROGRAMMING

We propose a dynamic programming (DP) based approach to efficiently solve the non-convex optimization problem formulated in Equation 3. Rather than optimizing the entire input time series simultaneously, the problem is decomposed into timestep-wise and token-wise subproblems, enabling sequential suboptimal solutions. This decomposition significantly improves computational tractability while preserving the effectiveness of the attack. The detailed procedure is presented in Algorithm 1. The output of the proposed method is an adversarial perturbation generated without any query to the target forecasting model; when applied to the raw input time series, this perturbation effectively manipulates the predictions of an LLM-based forecaster, causing substantial deviations from the original outputs.

### 4.4 DISCUSSION

The TDA is designed to exploit a previously under-explored attack surface in LLM-based time series forecasting: the publicly available tokenizer. Rather than repeatedly querying the target forecasting model, TDA operates by crafting small perturbations to raw time-series inputs that purposefully induce divergent token encodings. These misrepresentations then propagate through the model. By targeting tokenization rather than model inference, TDA sidesteps the query-based workflow used by many existing attacks and consequently reduces the risk of detection that arises from anomalous, repeated query traffic.

---

1: **Input:** Input time series $\mathbf{x}_t \in \mathbb{R}^T$, tokenization encoder $\mathcal{I}(\cdot)$ and decoder $\mathcal{I}'(\cdot)$, loss $\mathcal{L}_{\text{tokens}}(\cdot, \cdot)$, constraint $\epsilon$, vocabulary $\mathcal{N}$.

---

2: **for** $t = 1$ to $T$ **do**
3:     $S := \emptyset$
4:     $l := |\mathcal{N}|_0$
   # dynamic programming
5:     **for** $i = 1$ to length$(\mathcal{I}(\mathbf{x}_t))$ **do**
6:         Find $p_i$ (position of $\mathcal{I}(\mathbf{x}_t)_i$ in $\mathcal{N}$)
   # select the subset of vocabulary
7:         **if** $p_i > 0.5 * l$ **then**
8:             Select the left 30% of $\mathcal{N}$ as $\mathcal{N}'$
9:         **else**
10:        Select the right 30% of $\mathcal{N}$ as $\mathcal{N}'$
11:        **end if**
   # search adversarial examples from token space
12:        $\ell^* = \arg\max \mathcal{L}_{\text{tokens}}(\mathcal{I}(\mathbf{x}_t), S \cup \{\ell\})$
13:        s.t. $\|\mathbf{x}_t - \mathcal{I}'(S \cup \{\ell^*\})\|_p < \epsilon \,\&\, \ell \in \mathcal{N}'$
   # merge token$^{*(t)}$
14:        **if** $\ell^* \neq \emptyset$ **then**
15:           $S := S \cup \{\ell^*\}$
16:        **else**
17:           $S := S \cup \{\mathcal{I}(\mathbf{x}_t)_i\}$
18:        **end if**
19:     **end for**
   # compute temporal-level perturbations
20:     $\rho_t = \mathcal{I}'(S) - \mathbf{x}_t$
21: **end for**
   # collect perturbation sequences
22: **Return** $\boldsymbol{\rho} = \{\rho_t\}$ for $t \in [0, \ldots, T-1]$.

---

Algorithm 1: Perturbation $\boldsymbol{\rho}$ generation.

query-based workflow used by many existing attacks and consequently reduces the risk of detection that arises from anomalous, repeated query traffic.

TDA is explicitly engineered with real-world constraints in mind. The attack is executed **offline** (i.e., perturbations are computed locally using only the tokenizer) and injected into the data stream before the forecasting system consumes the data. Because no iterative queries to a cloud-based forecaster are required at attack time, TDA imposes minimal run-time overhead and can therefore be applied in latency-sensitive environments.

## 5 EXPERIMENT

In this section, we evaluate the effectiveness of TDA across six datasets and twelve baseline models, in comparison with two existing attacks. Our study focuses on two central questions: **Q.1** Does the proposed TDA significantly, stealthily, and efficiently degrade the predictive performance of existing LLM-based forecasters? **Q.2** What is the underlying mechanism by which TDA operates? Further details, including the computational costs of TDA and ablation studies of the DP-based solution, are provided in the Appendix E, F. Mitigation bypassing test and discussion are provided in Section 6.

**Baseline models.** We evaluate TDA against state-of-the-art forecasting systems, including ten LLM-based zero-shot forecasters and two non-LLM baselines. **Datasets.** Experiments are conducted on six real-world datasets: ETTh1, ETTh2, Traffic, Weather, Exchange, and Solar, covering diverse domains such as electricity, transportation, geoscience, economics, and energy. **Metrics.** We assess forecasting performance using two widely adopted error measures: Mean Absolute Error (MAE) and Mean Squared Error (MSE). **Baseline attacks.** For comparison, we include Gaussian White Noise (GWN) and a query-based, gradient-free targeted attack, Directional Gradient Approximation (DGA) (Liu et al., 2025). Detailed experimental settings are provided in Appendix D.

Table 1: Comparison of prediction performance with and without TDA across LLM-based and non-LLM-based forecasters. Lower MSE or MAE values indicate better forecasting accuracy. The worst and second-worst performance for each dataset-model combination are highlighted in bold and italics, respectively (only the worst performance is marked for non-LLM forecasters). Please note that the DGA requires continuous queries to the target model.

| Models | LLMTime w/ GPT-3.5 | | LLMTime w/ GPT-4 | | LLMTime w/ Llama 2 | | LLMTime w/ Mistral | | iTransformer (non-LLM) | | TimesNet (non-LLM) | |
|---|---|---|---|---|---|---|---|---|---|---|---|---|
| Metrics | MSE | MAE | MSE | MAE | MSE | MAE | MSE | MAE | MSE | MAE | MSE | MAE |
| ETTh1 | 0.073 | 0.213 | 0.071 | 0.202 | 0.086 | 0.244 | 0.097 | 0.274 | 0.071 | 0.218 | 0.073 | 0.202 |
| w/ GWN | 0.077 | 0.219 | 0.076 | 0.213 | 0.087 | 0.237 | 0.094 | 0.291 | 0.072 | 0.216 | 0.074 | 0.202 |
| w/ DGA | **0.085** | **0.249** | **0.083** | **0.232** | *0.091* | **0.251** | **0.098** | *0.295* | **0.075** | **0.226** | **0.081** | **0.213** |
| w/ TDA | *0.080* | *0.243* | *0.081* | *0.231* | **0.093** | *0.248* | **0.098** | **0.296** | 0.072 | 0.215 | 0.075 | 0.203 |
| ETTh2 | 0.263 | 0.372 | 0.155 | 0.267 | 0.237 | 0.373 | 0.277 | 0.492 | 0.171 | 0.296 | 0.166 | 0.316 |
| w/ GWN | 0.263 | 0.342 | 0.175 | 0.303 | 0.231 | *0.429* | 0.346 | 0.505 | 0.181 | 0.302 | 0.166 | 0.314 |
| w/ DGA | **0.275** | *0.408* | **0.201** | *0.327* | **0.257** | 0.425 | *0.356* | *0.554* | 0.179 | **0.308** | **0.169** | **0.321** |
| w/ TDA | *0.272* | **0.410** | *0.196* | **0.328** | *0.254* | **0.433** | **0.358** | **0.557** | **0.185** | 0.302 | 0.165 | 0.318 |
| Traffic | 0.837 | 0.844 | 0.805 | 0.779 | 0.891 | 1.005 | 0.826 | 0.973 | 1.081 | 0.995 | 1.095 | 1.022 |
| w/ GWN | 0.882 | 0.908 | 0.883 | 0.864 | 0.917 | 1.063 | 1.054 | 1.031 | **1.103** | 1.015 | 1.103 | 1.035 |
| w/ DGA | **0.955** | **1.073** | **1.417** | **1.214** | **0.994** | **1.083** | **1.744** | **1.217** | 1.097 | **1.034** | **1.155** | **1.047** |
| w/ TDA | *0.936* | *1.047* | *1.383* | *1.198* | *0.990* | *1.077* | *1.729* | *1.201* | 1.095 | 1.016 | 1.105 | 1.040 |
| Weather | 0.005 | 0.051 | 0.004 | 0.048 | 0.008 | 0.072 | 0.006 | 0.057 | 0.005 | 0.053 | 0.003 | 0.042 |
| w/ GWN | 0.005 | 0.053 | 0.005 | 0.051 | 0.008 | 0.074 | **0.007** | *0.066* | **0.006** | 0.063 | 0.003 | 0.042 |
| w/ DGA | **0.006** | *0.063* | **0.006** | *0.061* | **0.009** | 0.079 | 0.007 | 0.062 | **0.006** | **0.065** | **0.004** | **0.045** |
| w/ TDA | **0.006** | **0.065** | **0.006** | **0.070** | **0.009** | *0.078* | **0.007** | **0.068** | 0.005 | 0.054 | 0.003 | 0.043 |
| Exchange | 0.038 | 0.146 | 0.040 | 0.152 | 0.043 | 0.167 | 0.151 | 0.274 | 0.034 | 0.151 | 0.056 | 0.184 |
| w/ GWN | 0.042 | 0.179 | 0.046 | 0.182 | 0.050 | 0.185 | 0.160 | 0.298 | 0.044 | 0.166 | **0.065** | **0.195** |
| w/ DGA | *0.058* | **0.224** | **0.068** | *0.199* | **0.069** | **0.213** | **0.219** | **0.303** | **0.049** | **0.178** | 0.062 | 0.194 |
| w/ TDA | **0.060** | *0.221* | *0.066* | **0.201** | *0.065* | *0.202* | *0.208* | *0.301* | 0.045 | 0.168 | 0.064 | 0.190 |
| Solar | 0.316 | 0.325 | 0.235 | 0.276 | 0.297 | 0.304 | 0.303 | 0.314 | 0.233 | 0.262 | 0.301 | 0.319 |
| w/ GWN | 0.319 | 0.323 | 0.236 | 0.280 | 0.299 | 0.304 | 0.305 | 0.315 | 0.238 | 0.264 | 0.305 | 0.322 |
| w/ DGA | *0.342* | *0.355* | **0.291** | **0.308** | *0.315* | *0.318* | **0.327** | **0.346** | **0.240** | **0.277** | **0.317** | **0.330** |
| w/ TDA | **0.344** | **0.360** | *0.274* | *0.301* | **0.316** | **0.320** | *0.322* | *0.341* | 0.237 | 0.265 | 0.308 | 0.323 |

Table 2: Prediction performance of vanilla LLMs prompted as time series forecasters, reported with and without TDA. TDA consistently increases both MSE and MAE, demonstrating its disruptive impact on vanilla LLM forecasters.

| Models | ChatGPT 4o | | ChatGPT 4o-mini | | DeepSeek-R1 Llama-70B | | Llama 70B-Instruct | | Gemini 2.5 Flash | | Claude Sonnet 4 | |
|---|---|---|---|---|---|---|---|---|---|---|---|---|
| Metrics | MSE | MAE | MSE | MAE | MSE | MAE | MSE | MAE | MSE | MAE | MSE | MAE |
| Traffic | 1.231 | 0.956 | 1.236 | 1.080 | 1.408 | 1.142 | 1.503 | 1.274 | 1.235 | 1.122 | 1.229 | 1.035 |
| w/ TDA | **1.854** | **1.367** | **1.566** | **1.142** | **1.904** | **1.301** | **2.461** | **1.395** | **1.771** | **1.322** | **1.658** | **1.296** |
| Weather | 0.003 | 0.024 | 0.005 | 0.051 | 0.006 | 0.041 | 0.007 | 0.055 | 0.004 | 0.037 | 0.005 | 0.044 |
| w/ TDA | **0.005** | **0.043** | **0.007** | **0.062** | **0.008** | **0.053** | **0.008** | **0.061** | **0.006** | **0.052** | **0.007** | **0.057** |
| Solar | 0.294 | 0.311 | 0.308 | 0.317 | 0.302 | 0.319 | 0.307 | 0.324 | 0.299 | 0.322 | 0.312 | 0.316 |
| w/ TDA | **0.326** | **0.339** | **0.330** | **0.348** | **0.328** | **0.353** | **0.331** | **0.361** | **0.311** | **0.350** | **0.327** | **0.345** |

## 5.1 EFFECTIVENESS ANALYSIS (Q.1)

To evaluate the effectiveness and generalizability of the proposed TDA, we compare its performance with baseline attacks across a variety of forecasting models and datasets. Table 1 presents results on six models, including LLMTime (Gruver et al., 2024) with four LLM backbones (GPT-3.5, GPT-4, Llama 2, and Mistral), as well as two non-LLM forecasters, iTransformer (Liu et al., 2024b) and TimesNet (Wu et al., 2023), under both clean settings and three types of perturbations: GWN, DGA, and TDA. In addition, Table 2 reports the performance degradation caused by TDA on six vanilla and prompt-based LLM forecasters: ChatGPT-4o, ChatGPT-4o-mini, DeepSeek-R1-Distill-Llama-70B, Llama 70B-Instruct, Gemini 2.5 Flash, and Claude Sonnet 4.

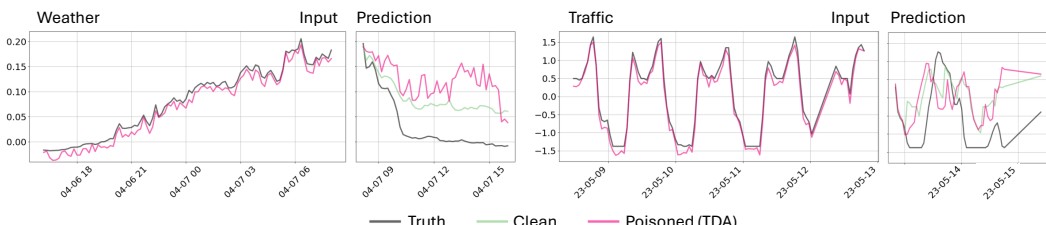

Figure 3: Comparison of LLMTime predictions using Mistral and GPT-4 backbones, with and without TDA, on the Traffic and Weather datasets.

Across all six datasets and a wide range of LLM-based forecasting models, TDA consistently induces severe performance degradation, surpassing GWN and achieving a disruptive power comparable to DGA, which requires iterative querying. On average, TDA increases MSE by 17% (ranging from 6% on ETTh1 to 43% on Exchange) and MAE by 13% (ranging from 6% on ETTh1 to 21% on Exchange) under a 2% perturbation budget. In contrast, non-LLM baselines such as iTransformer and TimesNet exhibit only marginal degradation under TDA, with results nearly indistinguishable from those under GWN. This contrast suggests that the vulnerability exploited by TDA primarily stems from the LLM tokenization interface, rather than from general model sensitivity to noise.

TDA also demonstrates greater efficiency compared to DGA. For example, on the Traffic dataset with LLMTime GPT-4, applying an attack with TDA requires only 5.8s, whereas DGA takes 14.7s (see Appendix E for details). We note that this comparison is not fully rigorous: TDA operates solely on a local tokenizer during perturbation generation, and its runtime can be further reduced with faster hardware. By contrast, DGA relies on repeated queries to a cloud-based LLM, where computational delay is subject to external factors such as network latency and API provider load. Taken together, TDA's efficiency and stealthiness underscore its practicality in real-world black-box settings.

### 5.2 INTERPRETATION STUDY (Q.2)

To gain deeper insights into the internal behavior of LLM-based forecasters under TDA, we conduct a qualitative interpretation study centered around the prediction trajectories and token-level deviations.

**Prediction Error and Input Bias.** Figure 3 illustrates the output deviation and input bias of LLMTime with different backbones (Mistral and GPT-4) on the Traffic and Weather datasets. The TDA curve exhibits a clear divergence from the ground truth, particularly at turning points and high-variability regions. This suggests that token-level disruptions significantly mislead the forecasting model, even when the overall perturbation magnitude is small.

**Latent Representations.** To investigate the impact of perturbations on internal representations, we employ t-distributed Stochastic Neighbor Embedding (t-SNE) to visualize the embeddings of both raw input sequences and their tokenized forms (Figure 4). In the temporal space, TDA-modified inputs remain nearly indistinguishable from clean data. However, in the token embedding space, they form clearly separable clusters, indicating that TDA induces nonlinear shifts in token representations. As a result, the LLM perceives perturbed inputs as semantically different from the originals, despite their numerical values being almost identical.

**Distributional Shifts.** In Figure 5, we quantify how TDA distorts the input, token, and output distributions. The input distribution under TDA is visually close to the clean distribution, consistent with the attack's low-magnitude constraint. However, token and output prediction distributions show clear rightward shifts, indicating semantic amplification of small numeric perturbations.

Together, these interpretation results demonstrate that token disruption propagates through the entire forecasting pipeline, from input bias to latent token mismatch to prediction error, validating the effectiveness and stealth of the proposed TDA.

## 6 ADVERSARIAL MITIGATION DISCUSSION

As demonstrated in our work, LLMs used for time series forecasting are highly susceptible to adversarial attacks, where even minimal perturbations can lead to significant prediction errors.

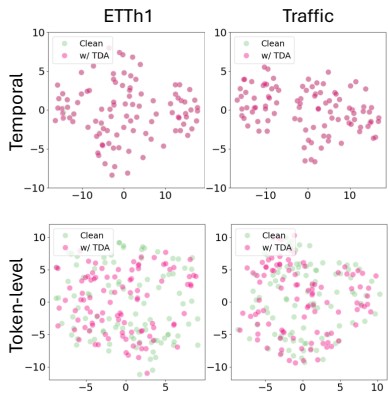

Figure 4: Temporal and token-level t-SNE analysis between clean and poisoned time series.

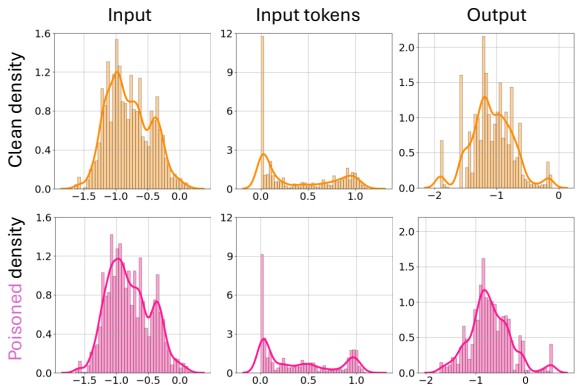

Figure 5: Comparison of input, encoded token, and prediction distributions for LLMTime w/ GPT-4 on the ETTh1 dataset under clean and TDA-poisoned inputs.

Although adversarial training (Shafahi et al., 2019) is an established method to improve model robustness by incorporating adversarial examples during training, it suffers from several critical drawbacks: it incurs a very high computational cost due to the need to generate and process vast numbers of adversarial samples, it is difficult to implement in black-box settings where model gradients and internal details are inaccessible, and it does not scale well in real-time applications or scenarios requiring frequent model updates. Considering the high cost of pre-training LLMs, traditional adversarial training methods are almost infeasible. Table 3 reports the effectiveness of common filter-based mitigation strategies (Xie et al., 2019) in countering TDA by measuring residual attack impact after preprocessing. The results show only marginal improvements, as such filters are unable to reshape the token-level distribution, leaving the forecasting models vulnerable.

As a cost-effective and scalable alternative, a watermark-based defense can be embedded directly into the tokenization process. Following Kirchenbauer et al. (2023), the vocabulary is partitioned into "green" and "red" lists using a random hash function, with a bias applied to favor green tokens. The expected proportion of green to-

Table 3: Filter-based mitigation bypassing tests.

|  |  | w/o defense | w/ Gaussian | w/ Mean |
|---|---|---|---|---|
| LLMTime Llama 2 | w/o attack | 0.244 | 0.247 | 0.246 |
|  | w/ TDA | 0.248 | 0.246 | 0.250 |
| LLMTime GPT-4 | w/o attack | 0.202 | 0.204 | 0.204 |
|  | w/ TDA | 0.227 | 0.226 | 0.223 |

kens follows a predictable distribution, and significant deviations, detected via a one-proportion $z$-test, may indicate adversarial manipulation. This approach integrates seamlessly without the need for expensive retraining. We aim to highlight a potential defense pathway that exploits the discrete token structure of LLM inputs, a perspective not yet explored in the context of time series forecasting and valuable for future research.

## 7 CONCLUSION

We propose Token Disruption Attack (TDA), a novel query-free adversarial method that targets the tokenization process in LLM-based time series forecasting. Unlike traditional attacks, TDA operates without model access by introducing subtle input perturbations that cause significant semantic shifts during tokenization.

Extensive experiments on 10 LLM-based forecasters across 6 applications demonstrate that TDA substantially degrades forecasting accuracy, with error increases exceeding 40% in some cases, while maintaining high stealth and efficiency. In contrast, non-LLM forecasters are largely unaffected, confirming that the vulnerability lies in the tokenization mechanism unique to LLMs.

These findings reveal a critical and previously underexplored attack surface in LLM-based forecasting pipelines. They further underscore the urgent need for research into tokenization-aware defenses to enhance the robustness and safety of next-generation forecasting systems.

## ETHICS STATEMENT

This work investigates the robustness of large language models in time series forecasting, which is critical in domains such as transportation, finance, and healthcare. Understanding and mitigating their vulnerability to adversarial attacks is essential for developing trustworthy AI systems. Our study provides insights into attack strategies and potential defenses, aiming to improve the safety and reliability of forecasting models in high-stakes settings. We are committed to ensuring our contributions are used responsibly to advance secure and robust AI technologies.

## REPRODUCIBILITY STATEMENT

We ensure reproducibility by fully describing the proposed Token Disruption Attack (TDA), including its formulation, optimization method, and evaluation setup. Experiments are conducted on publicly available datasets, and we specify both open-source baselines and commercial LLM APIs. For API-based LLMs, we detail the prompts and experimental settings. The implementation of TDA, along with preprocessing and evaluation scripts, is provided in the supplementary material and will be released publicly upon publication. Additional details, including computational costs, ablation studies, and mitigation bypassing tests, are included in the appendix to support transparent verification and future research.

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

# A  CASE STUDY

Table 4 presents a numerical toy case study. The clean input is $[1.02, 1.14, 0.91, 0.84, 0.78]$ and the true prediction result is 0.71. The predictions are generated by GPT-4o, following the same experimental settings used in Table 2 in the bold manuscript.

Table 4: Comparison of clean and poisoned data, along with corresponding tokens and predictions.

|          | Clean Data | Clean Tokens  | Poisoned Data | Poisoned Tokens |
|----------|-----------|---------------|---------------|-----------------|
| Input 0  | 1.02      | 16, 13, 3286  | 1             | 16              |
| Input 1  | 1.14      | 16, 13, 1265  | 1.131         | 16, 13, 16412   |
| Input 2  | 0.91      | 15, 13, 8956  | 0.9           | 16, 13, 24      |
| Input 3  | 0.84      | 15, 13, 9928  | 0.837         | 15, 13, 50016   |
| Input 4  | 0.78      | 15, 13, 4388  | 0.781         | 15, 13, 42872   |
| Pred.    | 0.72      | 15, 13, 8540  | 0.752         | 15, 13, 45157   |

Using Mean Absolute Error (MAE) as the evaluation metric, the case study reveals:

- The average perturbation to the inputs is only **0.0086**, indicating a very subtle modification.
- The prediction shifts from **0.72** to **0.752**, a difference of **0.032**.
- With respect to the true label (**0.71**), the original prediction error was **0.01**, which increases to **0.042** after the attack.
- This corresponds to a **4.2× amplification** in forecasting error.

This case study highlights the effectiveness of TDA in introducing minimal, stealthy perturbations while substantially degrading prediction accuracy, demonstrating its practicality as a no-query adversarial attack.

The toy case with clean input by the ChatGPT interface (anonymous link: clean), and the corresponding results are:

> **Conversation**
>
> **User:** Now you're a forecaster, and please output one number as the prediction. Your output should be one single number. Inputs: 1.02, 1.14, 0.91, 0.84, 0.78.
>
> **ChatGPT:** 0.72

The toy case with poisoned input by the ChatGPT interface (anonymous link: poisoned), and the corresponding results are:

> **Conversation**
>
> **User:** Now you're a forecaster, and please output one number as the prediction. Your output should be one single number. Inputs: 1., 1.131, 0.9, 0.837, 0.781.
>
> **ChatGPT:** 0.752

Additional reproductions with different prompts confirm similar results: example 1, example 2, example 3.

We acknowledge that this toy example may not generalize under all settings, especially when GPT-4o uses a step-by-step breakdown or tools. However, calling external tools or enabling CoT is not the focus of this paper since most LLM4TS models, e.g., LLMTime (Gruver et al., 2024) and TimeLLM (Jin et al., 2024), involve no use of these because of long input. In addition, a simpler forecasting task for the LLM inherently makes the attack more difficult. In the case study (5-step input and 1-step prediction), the model's high confidence and accuracy leave less room for disruption. This makes it harder for perturbations to significantly influence the output, especially when generalized across different prompts. Collectively, the result of this case study can still prove that a token-level attack can lead to a large prediction shift.

## B    Intuitive demonstration of Algorithm 1

A time series consists of many numeric values, each encoded into a sequence of tokens. The proposed DP-based algorithm constructs an adversarial version of a single number by modifying its token sequence while preserving its decoded numerical meaning. Repeating this process across all values yields the poisoned time series.

The algorithm operates token-by-token. For each token position, it first selects a subset of vocabulary candidates, intentionally choosing tokens from the opposite side of the vocabulary range to encourage impactful perturbations. Each candidate is then decoded back into numeric space. Candidates that maintain numerical closeness to the original value are retained, and among them, the algorithm selects the one that maximizes token-level deviation.

Validated tokens are incrementally merged with the partial sequence through a dynamic programming update, ensuring internal consistency. This process continues until all tokens for the number are processed. The result is a token sequence that remains numerically similar to the clean input but produces a meaningfully altered tokenization pattern capable of disrupting the forecasting model.

## C    Variables and Definitions

In this section, the meaning or definition of each variable is explained in detail in Table 5.

Table 5: Some important variables and their definitions.

| | |
|---|---|
| $d$ | The number of variables |
| $T$ | The length of historical input |
| $L$ | The length of future time series |
| $K$ | The total number of tokens per time step |
| $\tau$ | The number of whole tokens per time step |
| $\mathbf{x}_t$ | $d$-dimentional observations at time $t$ |
| $\mathbf{X}_t$ | A historical time series composed of $T$ observations |
| $\mathbf{Y}_t$ | A time series composed of observations in the next $L$ time steps |
| $\hat{\mathbf{Y}}_t$ | The prediction of future $L$ time steps |
| $f(\cdot)$ | The forecasting model |
| $\rho$ | The adversarial perturbation applied the clean historical time series |
| $\epsilon$ | The scale constraint of perturbations |
| $\mathcal{L}(\cdot, \cdot)$ | The loss function measuring the discrepancy between clean and poisoned prediction |
| $\mathcal{L}_{tokens}(\cdot, \cdot)$ | The loss function measuring the discrepancy between clean and poisoned tokenized representations. |
| $\mathcal{I}(\cdot)$ | The encode of the tokenizer |
| $\mathcal{I}'(\cdot)$ | The decode of the tokenizer |
| $\text{token}^{(t)}$ | Clean tokens at time $t$ |
| $\text{token}^{*(t)}$ | Poisoned tokens at time $t$ |
| $S$ | The candidate set of poisoned tokens at one time step |
| $\ell$ | Candidates of adversarial tokens |
| $\ell^*$ | The computed token maximizing the difference between clean and poisoned representatives |
| $\mathcal{N}$ | Vocabulary (token list) |

## D    Experiment Setup

We conducted experiments on diverse datasets to evaluate the effectiveness of TDA in disrupting LLM-based forecasting models. The experiments were performed on a system running Ubuntu 18.04 LTS, with PyTorch 1.7.1, Python 3.7.4, and an NVIDIA Tesla V100 GPU. The experimental procedure consisted of the following steps: (i) applying TDA to manipulate forecasts by poisoning the time series. (ii) using Gaussian White Noise (GWN) and Directional Gradient Approximation (DGA) as baselines for comparison, and (iii) assessing performance degradation using Mean Absolute Error (MAE) and Mean Squared Error (MSE). The perturbation scale, $\epsilon$, is set to 2% of the mean value of each dataset.

## D.1 BASELINE FORECASTING MODELS

**Vanilla LLMs**, including GPT-4o, GPT-4o-mini (Achiam et al., 2023), DeepSeek-R1-Distill-Llama-70B (Liu et al., 2024a), Llama 70B-Instruct (Touvron et al., 2023), Gemini 2.5 Flash, and Claude Sonnet 4, by prompting them as forecasters. It should be noted that long-term forecasting is formulated as an iterative process, where each predicted output is recursively fed back into the model to generate the next-step prediction.

> **Prompt-based forecaster**
>
> **System:** You are a time series forecaster. The user will provide a time series as input, and your task is to predict the next value. Output only a single numerical value, without any explanation or additional text.
>
> **User:** Please output one number as the prediction. Your output should be one single number. Inputs: ...
>
> **User (format check):** Your previous output format is incorrect. Please return only one numerical value as the prediction, with no additional text, symbols, or explanation. Inputs: ...

**LLMTime** (Gruver et al., 2024): This model frames time series forecasting as a next-token prediction task, leveraging LLM architectures such as GPT and Llama. By encoding time series data into numerical sequences, LLMTime enables these models to apply their sequence modeling capabilities to forecasting. To evaluate the robustness of adversarial attacks, we test LLMTime with base models including GPT-3.5, GPT-4, Llama 2, and Mistral, analyzing their resilience when adapted from natural language processing to time series forecasting.

**TimesNet** (Wu et al., 2023) and **iTransformer** (Liu et al., 2024b), two transformer-based models designed to capture long-term dependencies in time series data, are included to assess the performance and robustness of LLM-based forecasting models against non-LLM approaches.

## D.2 BASELINE ATTACK

**Gaussian White Noise** (GWN) serves as a naive baseline, where random noise is added to the input time series without any optimization. This allows us to assess whether the performance degradation caused by TDA is due to a structured attack rather than mere stochastic perturbations.

**Directional Gradient Approximation** (DGA) (Liu et al., 2025) is a gradient-free attack method that queries the target model to estimate gradients and compute adversarial perturbations. Since DGA requires continuous interaction with the model, comparing it with TDA, which operates in a strictly query-free setting, demonstrates the unique advantage of disrupting the tokenization process without direct model access.

## D.3 DATASETS

We conduct experiments on four datasets, representing distinct forecasting challenges.

The **ETTh1** and **ETTh2** dataset (Zhou et al., 2021) consists of hourly temperature and power consumption measurements from electricity transformers over a two-year period. This dataset captures both short-term fluctuations and long-term seasonal trends, making it a valuable benchmark for evaluating forecasting performance.

The **Traffic** dataset (Gruver et al., 2024) provides hourly traffic volume data from Istanbul, reflecting the complex temporal dependencies influenced by congestion cycles and road network dynamics. Due to its highly volatile nature, it poses significant challenges for time series forecasting models.

The **Weather** dataset (Zeng et al., 2023) includes hourly meteorological observations, such as temperature, humidity, and wind speed. The inherent variability and nonlinear dependencies in weather conditions make forecasting particularly difficult, requiring models to capture both short-term fluctuations and broader climatic trends.

The **Exchange Rates** dataset (Lai et al., 2018) comprises daily foreign exchange rate data for eight different countries, covering the period from 1990 to 2016. This dataset provides insight into long-term economic trends and complex temporal dependencies in global financial markets.

The **Solar** dataset (Lai et al., 2018) contains solar power production data collected in 2006, sampled every 10 minutes from 137 photovoltaic (PV) plants in the state of Alabama. This dataset captures fine-grained temporal dynamics of renewable energy generation and reflects variability driven by environmental and weather-related factors.

Table 6: Detailed dataset descriptions.

| Dataset | Dim | Frequency | Size | Information |
|---------|-----|-----------|-------|----------------|
| ETTh1   | 7   | Hourly    | 14307 | Electricity    |
| ETTh2   | 7   | Hourly    | 14307 | Electricity    |
| Traffic | 1   | Hourly    | 5310  | Transportation |
| Weather | 21  | 10 minute | 52603 | Geoscience     |
| Exchange| 8   | Daily     | 7207  | Economy        |
| Solar   | 137 | Hourly    | 52179 | Energy         |

For all datasets, we adopt a standardized data split, allocating 60% for training, 20% for validation, and 20% for testing. All forecasting models are trained with a 96-step historical window to predict the next 48 time steps, ensuring consistency across experimental settings.

### D.4    METRICS

Mean Absolute Error (MAE) and Mean Squared Error (MSE) are employed to evaluate the performance of LLM-based forecasting models as well as the effectiveness of adversarial attacks. Given the ground truth $\mathbf{Y}_t$ and the prediction $\hat{\mathbf{Y}}_t$, they are defined as:

$$\text{MAE} = \frac{1}{T} \sum_{t=1}^{T} \left| \mathbf{Y}_t - \hat{\mathbf{Y}}_t \right|, \tag{5}$$

$$\text{MSE} = \frac{1}{T} \sum_{t=1}^{T} \left( \mathbf{Y}_t - \hat{\mathbf{Y}}_t \right)^2, \tag{6}$$

where $T$ denotes the forecasting horizon.

### D.5    PREDICTION ERROR AND INPUT BIAS

Figure 6 illustrates the output deviation and input bias of LLMTime with different backbones (Mistral and GPT-4) on the Traffic and Weather datasets. The TDA curve exhibits a clear divergence from the ground truth, particularly at turning points and high-variability regions. This suggests that token-level disruptions significantly mislead the forecasting model, even when the overall perturbation magnitude is small. Under normal conditions, the predicted trajectory closely follows the ground truth, while the TDA curve exhibits a clear divergence from the ground truth, particularly at turning points and high-variability regions. This suggests that token-level disruptions significantly mislead the forecasting model, even when the overall perturbation magnitude is small.

We further analyze the input bias and prediction error over time, which show that TDA leads to not only larger bias but also persistent accumulation of forecasting errors. This effect is more prominent in high-variability domains like traffic flow, where even subtle misrepresentations can cascade into substantial forecasting deviations.

## E    COMPUTATIONAL COST EVALUATION

The proposed TDA replaces cloud-based queries with local tokenizer queries. Only the user submits the (potentially poisoned) time series to the cloud-based LLM forecaster independently, and the

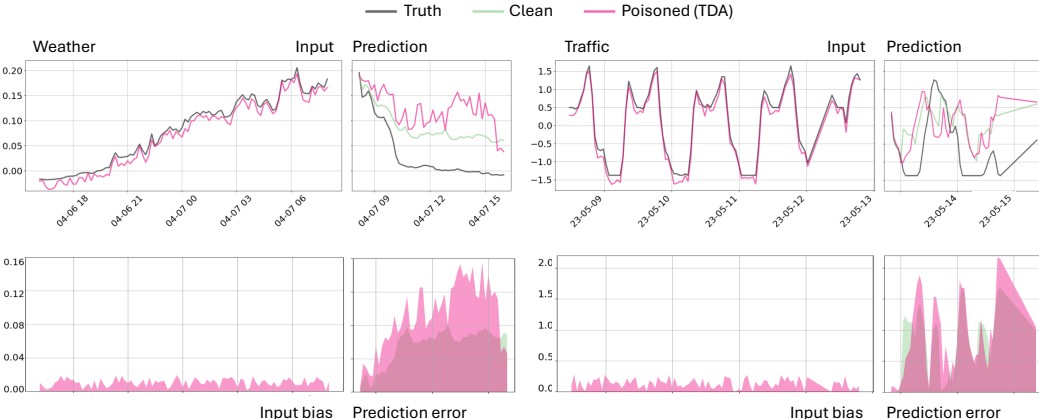

Figure 6: Comparison of LLMTime predictions using Mistral and GPT-4 backbones, with and without TDA, on the Traffic and Weather datasets. The figure demonstrates the stronger disruptive effect of TDA, leading to greater deviations from the ground truth.

attacker can manipulate the forecasting results without issuing any direct queries to the LLM-based forecaster on the cloud, making the attack both stealthier and more practical.

We provide a running time comparison between TDA and DGA (Liu et al., 2025). In this experiment, we target the LLMTime with GPT-4 across two datasets. The comparison is recorded in Table 7. The results suggest that the proposed method is more efficient:

Table 7: Comparison of computational time between DGA and the proposed TDA.

|  | DGA | Proposed |
|---|---|---|
| LLMTime GPT-4, Weather | 214.4s | **37.6s** |
| LLMTime GPT-4, Solar | 1077.5s | **227.4s** |
| LLMTime GPT-4, Exchange | 52.4s | **18.3s** |
| LLMTime GPT-4, Traffic | 14.7s | **5.8s** |

However, we acknowledge that this experiment is not fully rigorous, as the proposed TDA only queries the local tokenizer during perturbation generation. This local process can be easily accelerated using better computational hardware. In contrast, DGA relies on iterative queries to a cloud-based LLM, where the computational delay is subject to external factors such as internet speed and API provider's load—both are unpredictable and not under the attacker's control.

Here is the comparison of the pipeline of query-based (DGA) and query-free (TDA) attacks.

**DGA:** Query the cloud-based forecaster $\longrightarrow$ Generate perturbations $\longrightarrow$ Poison data

**TDA:** Query local tokenizer $\longrightarrow$ Generate perturbations $\longrightarrow$ Poison data

We emphasize that the key advantage of our method lies in its novel no-query mechanism, which eliminates the need for iterative and uncontrollable cloud queries, a limitation that significantly affects the practicality of previous methods.

## F  ABLATION STUDIES ON DP-BASED SOLUTION

Our primary contribution lies in proposing a novel no-query attack mechanism tailored for LLM-based time series forecasting. To operationalize this idea, we designed a dynamic programming (DP)-based solver to handle the token-level perturbation process under cardinality constraints. This solver provides an effective way to approximate a challenging non-convex, NP-hard problem within a reasonable time and computational budget. The computational complexity of the proposed DP-based

solution is $\mathcal{O}(l \times T)$, where $l$ is the length of the vocabulary and $T$ is the length of the historical input.

This section benchmarks the proposed DP-based solution against alternative solvers to evaluate both time complexity and solution quality. We consider two baselines:

- **DP-based solution without vocabulary subset selection.** This variant follows Algorithm 1 but omits the vocabulary subset selection step (lines 7–11). It serves as an ablation study to assess the benefit of this component.
- **Genetic algorithm-based solution.** This baseline employs an evolutionary strategy, detailed as follows:
  - **Initialization:** Randomly generate 50 tokens from the token table as the initial parent population.
  - **Crossover:** Randomly select two parents and compute the mean of their values to form a child token. Repeat 50 times to produce 50 children.
  - **Merge:** Combine the 50 parents and 50 children into a pool of 100 candidates.
  - **Selection:** Choose the 50 tokens with the largest difference from the target (clean) token, subject to numerical constraints.
  - **Iteration:** Use the selected tokens as the new parent population and repeat the process for 50 iterations.

The experiments are conducted on the ETTh1 dataset, representing a power forecasting application, and performance is evaluated using the MSE metric. Table 8 summarizes the comparison results. The proposed DP-based solver achieves superior effectiveness and stability relative to the genetic algorithm baseline, while also being more computationally efficient than both the genetic algorithm and the DP variant without vocabulary subset selection. Overall, the results demonstrate that the proposed DP-based solution is an effective, efficient, and robust approach for generating token-level perturbations.

Table 8: comparative study benchmarking the proposed DP-based solver against alternatives.

| | **Proposed** | **DP w/o subset** | **Genetic** |
|---|---|---|---|
| LLMTime Llama 2 | 0.086 | 0.086 | 0.086 |
| LLMTime Llama 2 w/ TDA | 0.093 | 0.094 | 0.089 |
| $\Delta$ | *8.1%* | **9.3%** | 3.5% |
| Time (s) | **30.6** | 79.2 | 127.8 |
| LLMTime GPT-4 | 0.071 | 0.071 | 0.071 |
| LLMTime GPT-4 w/ TDA | 0.081 | 0.080 | 0.080 |
| $\Delta$ | **14.1%** | *12.7%* | *12.7%* |
| Time (s) | **35.2** | 88.4 | 144.3 |

We note that the proposed DP-based method is not guaranteed to be optimal in terms of solution quality or efficiency when compared with other possible solvers. However, since the primary goal of this work is to demonstrate the feasibility and impact of token-level adversarial attacks under realistic constraints (e.g., no-query, black-box settings), the DP-based approach should be regarded as a practical starting point that highlights the tractability and effectiveness of the proposed threat model.

# G  ATTACK EFFECTIVENESS UNDER TOKENIZER MISMATCH

In some corner cases or future forecasting services, mismatched tokenizers may arise depending on the user tier or request properties. This section evaluates how such tokenizer mismatches affect attack performance.

We conduct several transfer-based attacks in which perturbations are generated using the tokenizers of ChatGPT 4o and Gemini 2.5 Flash. These perturbations are then applied to four different LLM-based forecasters, as reported in Table 9. The results show that adversarial perturbations produced from mismatched tokenizers yield, on average, a 57% smaller error drop compared to the original TDA.

Table 9: Attack effectiveness evaluation on mismatched tokenizers.

| Models | LLMTime w/ GPT-3.5 | | LLMTime w/ GPT-4 | | LLMTime w/ Llama 2 | | LLMTime w/ Mistral | |
|---|---|---|---|---|---|---|---|---|
| Metrics | MSE | MAE | MSE | MAE | MSE | MAE | MSE | MAE |
| Traffic | 0.837 | 0.844 | 0.805 | 0.779 | 0.891 | 1.005 | 0.826 | 0.973 |
| w/ TDA$_{\text{ChatGPT 4o}}$ | 0.884 | 0.915 | 0.922 | 0.917 | 0.893 | 1.024 | 1.051 | 1.039 |
| w/ TDA$_{\text{Gemini 2.5 Flash}}$ | 0.867 | 0.906 | 0.898 | 0.905 | 0.916 | 1.033 | 1.118 | 1.054 |
| w/ TDA$_{\text{original}}$ | **0.936** | **1.047** | **1.383** | **1.198** | **0.990** | **1.077** | **1.729** | **1.201** |
| Exchange | 0.038 | 0.146 | 0.040 | 0.152 | 0.043 | 0.167 | 0.151 | 0.274 |
| w/ TDA$_{\text{ChatGPT 4o}}$ | 0.045 | 0.187 | 0.051 | 0.177 | 0.050 | 0.188 | 0.159 | 0.281 |
| w/ TDA$_{\text{Gemini 2.5 Flash}}$ | 0.044 | 0.181 | 0.048 | 0.169 | 0.049 | 0.176 | 0.189 | 0.293 |
| w/ TDA$_{\text{Original}}$ | **0.060** | **0.221** | **0.066** | **0.201** | **0.065** | **0.202** | **0.208** | **0.301** |

## H EFFECTIVENESS EVALUATION ON VARYING INPUT/OUTPUT LENGTH

This section evaluates the attack effectiveness under varying input and output lengths, beyond the default 96-input and 48-output setting. MAE is used as the evaluation metric. The results are summarized in Table 10. Our experimental results highlight several important observations. First, the proposed TDA consistently degrades the performance of LLM-based forecasting models, demonstrating its robustness even under very long prediction horizons.

Table 10: Forecasting performance on Traffic dataset under varying horizons, with and without TDA.

| Models | Traffic 48 | Traffic 168 | Traffic 336 | Traffic 720 | Traffic 1024 |
|---|---|---|---|---|---|
| LLMTime (w/o TDA) | 0.844 | 0.907 | 0.972 | 1.161 | 1.235 |
| LLMTime (w/ TDA) | 1.047 | 1.133 | 1.204 | 1.305 | 1.398 |

## I LLM USAGE STATEMENT

We used ChatGPT-5 exclusively for grammar checking and editorial improvements to the manuscript's prose.

All scientific contributions, specifically the conception and formulation of the Token Disruption Attack (TDA), algorithm development, experiment design and execution, data analysis, and interpretation of results, are original and were performed by the authors.

Large Language Models (LLMs) play a central role in this work only as *targets* of our adversarial evaluation (i.e., they serve as the forecasting systems we analyze and attack); their use as experimental subjects is thoroughly described in Section 5 for experiment settings.

No LLM was used to generate research hypotheses, design the attack, run experiments, or search related works.

