# OpenReview forum: "Manipulate Large Language Models in Time Series Forecasting by Token Disruption"
_ICLR.cc/2026/Conference — Submitted to ICLR 2026_

### Official Review · Reviewer_1qYA · 2025-10-27

**Soundness:** 3
**Presentation:** 3
**Contribution:** 3
**Rating:** 6
**Confidence:** 3

**Summary:**

This paper tackles the vulnerability of LLMs as time series forecasting, where a subtle perturbations in raw time series can induce severe forecasting errors. Since existing attacks typically rely on repeated queries which are easily detectable, the authors introduce the Token Disruption Attack (TDA) that generates perturbations by solely querying the local tokenizer which is publicly available. The authors then formulate TDA as a non-convex optimization problem and design a dynamic programming-based method to solve it efficiently. Extensive experiments on 10 LLM-based forecasters across 6 applications demonstrate that TDA substantially degrades forecasting accuracy, while maintaining high stealth and efficiency.

**Strengths:**

1. **Originality of research question**: The authors clearly articulate the limitations of existing attacks -- iterative queris to the target forecasting model incurs high computational costs and makes them easily detectable. Building on this, they propose an original and well-defined research question with clear constraint -- manipulate an attack without querying the target model.
2. **Innovative method design**: Despite the strong constraint of operating without querying the target model, the authors leverage the publicly available tokenizer, and reformulates the attack in tokenization process.
3. **Extensive experiments and analysis**: The authors execute experiments on 10 LLM-based forecasters across 6 applications, and provide detailed quantitative and qualitative analyses. For example, the demonstration of imperceptible perturbation (Figure 3) is clear, we can observe that the input signals remain close, yet the outputs diverge significantly.

**Weaknesses:**

1. Although the overall method design is innovative, the technical description lacks sufficient depth, particularly for the dynamic programming–based perturbation generation presented in Section 4.3. The current form of Algorithm 1 is difficult to follow, and several key computational steps are not clearly explained in the main text. It is recommended that the authors provide additional descriptions or illustrative explanations of the algorithm’s critical steps to improve readability.
2. Some figures are not fully clear or consistent with the accompanying discussion. For example, in Figure 4, the clean sample points appear to be missing in the temporal space visualization. In Figure 5, while I agree that the input distributions are visually similar, I do not observe the rightward shift of input token distributions claimed in line 423. The authors may consider revising these figures or clarifying their visual interpretation to ensure consistency between text and visualization.

**Questions:**

1. Could the authors elaborate on the intuition behind the DP decomposition and how it guarantees effective perturbation search?
2. In Figure 4, the clean samples seem missing in the temporal space visualization. Could the authors clarify whether this omission was intentional or due to visual overlap?
3. In Figure 5, the text (line 423) mentions a rightward shift in the input token distribution under TDA, but this shift is not visually apparent. Could the authors explain how this shift is quantified, or revise the figure to better illustrate the described phenomenon?

---

> ### Author Response · Authors · 2025-12-03
>
> Thanks so much for your recognition, constructive reviews, and sincere comments. We address each point in detail below.
>
> ---
>
> ### **1. The technical description lacks sufficient depth**
>
> #### **Could the authors elaborate on the intuition behind the DP decomposition and how it guarantees effective perturbation search?**
>
> We appreciate this insightful comment. We agree that Section 4.3 was overly simplified and did not fully convey how the proposed DP-based algorithm operates.
>
> Below is a clearer explanation of the intuition and workflow.
>
> **Intuition**
>
> A time series consists of a long sequence of numeric values, and each value can be encoded into a sequence of tokens. The proposed DP-based algorithm generates an adversarial example for a single number; by running this procedure iteratively across all numbers in the sequence, we obtain the fully poisoned time series.
>
> The DP-based algorithm constructs the poisoned token sequence **token-by-token**, ensuring that each intermediate candidate satisfies the numerical similarity constraints while guiding the perturbation toward impactful regions of the tokenizer space.
>
> **Step-by-step process**
>
> 1. **Vocabulary subset selection**
>    For each token, we first identify a meaningful subset of the vocabulary. If the numeric token lies on the **left** side of the vocabulary range, we select poisoned candidates from the **right** side, and vice versa.
>
> 2. **Constraint checking via decoding**
>    For each candidate token, we decode the partially formed token sequence back into numeric space.
>    - If the decoded value remains **numerically close** to the clean value, we keep the candidate.
>    - Otherwise, we fall back to the original clean token.
>    - From the saved candidates, we pick the one that maximises the token-level difference.
>
>    This ensures that **numerical similarity constraints** are always satisfied.
>
> 3. **Merging with previous steps**
>    After validating a poisoned token, we merge it with the previously constructed partial sequence.
>    This dynamic programming step preserves consistency while enabling stepwise refinement.
>
> 4. **Iterative refinement**
>    We repeat this process until all tokens representing a number have been processed.
>    The final result is a poisoned token sequence that:
>    - Respects numerical closeness,
>    - Follows a structured search pattern, and
>    - Produces an output that meaningfully disrupts the output of the tokenization process.
>
> Thank you again for highlighting the need for additional clarification. We add a very detailed version to **Appendix B**.
>
> ---
>
> ### **2. Some figures are not fully clear**
>
> #### **2.1 Missing clean samples in Figure 4**
>
> It is caused by **visual overlap**.
>
> Figure 4 illustrates the difference in the temporal dimension between clean and poisoned samples. When zoomed in, small green circles (clean tokens) appear slightly shifted from the pink circles (poisoned tokens).
>
> This figure demonstrates that TDA produces perturbations that:
> - meaningfully alter the **tokenization output**,
> - while preserving **numerical similarity**.
>
> ---
>
> #### **2.2 Rightward shift in Figure 5 is not visually clear**
>
> Thank you for pointing this out. We intended to describe how the poisoned tokens become **more concentrated toward the right side** of the distribution (i.e., closer to 1.0) compared to the clean tokens.
>
> To avoid ambiguity, we will revise the wording to:
>
> > “The poisoned tokens become more rightward-concentrated.”
>
> This phrasing more accurately reflects the intended observation and avoids misinterpretation.
>
> ---
>
> We appreciate the reviewer’s helpful suggestions, which significantly improve the clarity and presentation of our work.

---

### Official Review · Reviewer_6SaQ · 2025-10-28

**Soundness:** 3
**Presentation:** 2
**Contribution:** 2
**Rating:** 4
**Confidence:** 3

**Summary:**

This paper studies query-free attacks on LLM-based time series forecasters by perturbing inputs so that, after tokenization, the token encodings diverge substantially from the clean ones. The authors formalize a non-convex objective over token differences (considering temporal patterns, decimals, delimiters) and propose a dynamic-programming (DP)–based solver to select adversarial tokens under a budget constraint. Experiments across multiple LLM forecasters and datasets show 7–41% error increases with small perturbation budgets, while non-LLM forecasters are reported as largely unaffected.

**Strengths:**

[1] Introduces a novel attack surface by manipulating tokenization without querying the model.
[2] Dynamic programming heuristic is a creative method for non-convex search under budget constraints.
[3] Covers a diverse set of datasets and models, showing broad effectiveness.
[4] The paper provides insightful qualitative analyses, including embedding visualization, token alignment, and uncertainty distribution.

**Weaknesses:**

[1] In Algorithm 1, how are the candidate tokens N restricted in practice? This paper lacks clear interpretation and justification.
[2] Although the paper targets a novel attack surface, comparisons to adversarial methods on time series would contextualize the novelty and strength of the approach.
[3] Some replicated definition, define X_t \in R^{d \times T} at line 161 but X_t \in R^{T}  in Algorithm. This cause confusion.
[4] Writing errors, e.g Table header spells “Metrcis” instead of “Metrics”;
[5] There are duplicated “REFERENCES” section . A full References block appears (Page 10), then another References block starts again in Page 13 at Line 654.
[6] “Equation. 2” has an extra period (Line 192); use “Eq. (2)” or “Equation (2)”.

**Questions:**

[1] Since TDA operates through a local tokenizer, how transferable is the attack across different tokenization schemes
[2] The dynamic programming solver is claimed to be efficient, but the vocabulary size in LLM tokenizers is typically large. How is the candidate subset selected?
[3] All experiments use a 96→48 forecasting setup. How does the attack perform with longer horizons or variable-length input sequences?
[4] To what extent do formatting choices (e.g., token separators, decimal formatting, prompt templates) influence attack success? Could prompt engineering mitigate the attack impact?

---

> ### Author Response · Authors · 2025-12-03
>
> Thanks so much for your time and sincere comments. We revised the submission based on your suggestions, which significantly improved the paper.
>
> ---
>
> ### **1. Restriction of candidate tokens in Algorithm 1 (weakness 1)**
>
> The practical restriction is implemented through Equation (3) and Line 13 of Algorithm 1.
>
> For each candidate token, we apply the decoder $\mathcal{I}'$ to map it from token space back to numerical space, then compute its deviation from the clean numeric value. The candidate is retained only if this deviation is small; otherwise, the algorithm keeps the original clean token. This ensures that all selected candidates satisfy the numerical similarity constraint.
>
> ---
>
> ### **2. Comparison with adversarial attacks on time series (weakness 2)**
>
> We use DGA [1], a black-box, query-based attack, as a baseline for comparison.
>
> Existing adversarial attacks on non-LLM forecasting models are predominantly white-box methods [2–4], whereas LLM-based forecasters must be handled under strict black-box constraints. Lines 139–144 in the Related Work section discuss these distinctions.
>
> ---
>
> ### **3. Inconsistent definitions of $X_t$ (weakness 3)**
>
> Thank you for pointing this out. In Algorithm 1, it should indeed be $x_t \in \mathbb{R}^{T}$.
>
> LLM-based forecasters treat multivariate forecasting as multiple independent univariate forecasting tasks. Thus, Algorithm 1 describes the perturbation generation process for a single variable, and the multivariate case is handled by running the procedure multiple times.
>
> ---
>
> ### **4. Writing errors (weaknesses 4, 5, 6)**
>
> We appreciate the detailed corrections. These issues have been fixed in the revised manuscript, resulting in improved clarity and readability.
>
> ---
>
> ### **5. Transferability across tokenization schemes (question 1)**
>
> In response, we added an empirical study on attack effectiveness under tokenizer mismatch in **Appendix G**. We conduct transfer-based attacks in which perturbations are generated using the tokenizers of ChatGPT 4o and Gemini 2.5 Flash, and then applied to four different LLM-based forecasters. The results show that adversarial perturbations generated from mismatched tokenizers lead to, on average, a 57% smaller error drop compared to the original TDA.
>
> ---
>
> ### **6. Candidate subset selection under large vocabulary sizes (question 2)**
>
> This is an excellent question highlighting an important distinction between LLMs and LLM-based forecasters.
>
> Using GPT-4 as an example:
> - Total tokenizer vocabulary: **100,277** tokens
> - Number of tokens representing numeric content: **1,128** tokens
>
> Thus, although the entire vocabulary is large, the **numeric sub-vocabulary is only ~1%** of the total. This makes DP-based search computationally feasible.
>
> For candidate selection, if the original numeric token lies on the left side of the numeric vocabulary, we select candidates from the right side, and vice versa. This directional search increases the likelihood of producing impactful perturbations.
>
> ---
>
> ### **7. Performance under longer horizons or variable-length inputs (question 3)**
>
> **Appendix H** evaluates the attack effectiveness under varying input and output lengths, beyond the default 96-input and 48-output configuration. We observe that:
> 1. The proposed TDA consistently degrades forecasting performance across all settings; however,
> 2. Its attack effectiveness decreases as the output horizon becomes longer.
>
>
> ---
>
> ### **8. Impact of formatting choices such as token separators and prompt templates (question 4)**
>
> Formatting does influence attack behavior, especially in toy experiments. Examples are reproduced here:
> [link 1](https://chatgpt.com/share/6845be2a-06c4-8003-89a8-cb903e17f362)
> [link 2](https://chatgpt.com/share/68d349f0-72d8-8003-bd2f-ff9d514529c4)
>
> For example, adding “[” or “]” around numbers in toy settings can invalidate the attack because modern LLMs internally invoke auxiliary agents that change how tokenization is handled.
>
> However, this issue does not arise for LLM-based forecasters such as Time-LLM [5], where prompt templates and input formats are standardized and fixed by the forecasting framework. This is why formatting sensitivity is not the focus of our study.
>
> We fully agree that designing **prompt-based adversarial defenses** is an interesting direction for future work.
>
> ---
>
> ### **References**
>
> [1] *Adversarial Vulnerabilities in Large Language Models for Time Series Forecasting*, AISTATS 2025
> [2] *Adversarial Attacks on Probabilistic Autoregressive Forecasting Models*, ICML 2020
> [3] *Practical Adversarial Attacks on Spatiotemporal Traffic Forecasting Models*, NeurIPS 2022
> [4] *Robust Multivariate Time-Series Forecasting: Adversarial Attacks and Defense Mechanisms*, ICLR 2023
> [5] *Time-LLM: Time Series Forecasting by Reprogramming Large Language Models*, ICLR 2024

---

### Official Review · Reviewer_GxMJ · 2025-10-31

**Soundness:** 3
**Presentation:** 2
**Contribution:** 2
**Rating:** 2
**Confidence:** 4

**Summary:**

This paper proposes a **Token Disruption Attack (TDA)**, a novel *query-free* adversarial method targeting large language models (LLMs) used for time series forecasting. TDA perturbs the **tokenization process** itself by introducing small modifications to the raw time series before tokenization. The method formulates this as a non-convex optimization problem that maximizes the divergence between tokenized representations of clean and perturbed sequences, solved heuristically via a **dynamic programming–based algorithm**.

**Strengths:**

(1) TDA is a query-free attack. This design improves stealthiness and computational efficiency, which are relevant considerations in real-world forecasting scenarios.

(2) The experimental section evaluates the proposed method on multiple LLM backbones (e.g., GPT-3.5/4, Llama-2, Mistral, Gemini, Claude) and diverse datasets (e.g., ETTh, Traffic, Weather, Solar), demonstrating that small perturbations can substantially degrade model performance.

(3) As LLMs increasingly appear in time series applications (e.g., TimeGPT, Chronos, Time-LLM), exploring their adversarial robustness is a relevant and timely problem for the ICLR community.

**Weaknesses:**

While the paper presents an interesting perspective on query-free attacks against LLM-based forecasters, several issues limit its clarity, rigor, and overall contribution.

**(1) Ambiguous threat model.** The proposed attack assumes access to a “local tokenizer,” yet the paper fails to articulate any realistic setting in which an adversary could both control the local tokenizer and benefit from degraded forecasts. In practice, only the model provider (like OpenAI) or the end user would typically have access to the tokenizer, and neither party has an incentive to deliberately attack the model to obtain inaccurate results. This makes the attack scenario difficult to justify in real-world contexts.

**(2) Poorly defined figures and metrics.** Section 4.2 introduces the concepts of “first-level” and “second-level” alignment without explicit labeling or quantitative definitions. Moreover, the mathematical exposition surrounding Eq. (4) is incomplete: the specific norm used is not identified, and the distinct roles of $token_i$ (whole tokens) and $token_j$ (fractional tokens) remain unclear. From my perspective, these two types of tokens appear to measure similar differences, so further explanation is needed to clarify how this index distinction meaningfully affects their functional roles.

**(3) Weak algorithmic explanation.** Section 4.3 presents pseudocode but provides almost no intuitive description of how the dynamic-programming procedure accomplishes its optimization goal, nor why it represents a reasonable relaxation of the non-convex problem in Eq. (3). As a result, the algorithm feels more heuristic than principled.

**(4) Lack of theoretical or empirical justification.** The paper offers no theoretical guarantee to explain why the proposed procedure could effectively disrupt forecasting outputs. Without theoretical support, the contribution (particularly concerning the role of the dynamic-programming design) remains largely empirical.

**(5) Experimental concerns.** The experimental results show that TDA performs comparably or even worse than DGA on several datasets (e.g., *ETTh1* and *Traffic*), where DGA outperforms. These mixed outcomes weaken the claimed superiority of TDA and suggest that the proposed method may not provide consistent advantages across datasets or model families.

**Questions:**

See the weakness

---

> ### Author Response · Authors · 2025-12-03
>
> Thanks for your time and reviews. We respectfully clarify that weaknesses **1, 2, and 5** arise from misunderstandings, while weaknesses **3 and 4** reflect common unsolved challenges in NP-hard problems, adversarial attacks and LLM research.
>
> ---
>
> ### **1. Ambiguous threat model**
>
> Reviewer GxMJ argues that the proposed threat model is unrealistic based on the opinion that *“only the model provider (like OpenAI) or the end user would typically have access to the tokenizer.”*
>
> We respectfully clarify that this opinion is **Factually Incorrect**. In practice, anyone can easily access the tokenizer locally with just a few lines of code. Almost all commercial LLMs publicly release their tokenizers, and numerous community tools, such as **Tiktokenizer**, aggregate tokenizers for a wide range of models locally.
>
> For example, obtaining OpenAI’s tokenizer is straightforward, without any API key:
>
> ```
> import tiktoken
>
> model = "gpt-4" # To get the tokeniser corresponding to a specific model
> enc = tiktoken.encoding_for_model(model)
> tokens = enc.encode('input')
> ```
> Importantly, accessing the tokenizer is **far easier** than querying a cloud-based LLM-forecasters, as it requires no privileged permissions, no model weights, and no API computation. This strongly supports the feasibility and realism of the proposed query-free, tokenization disruption attack.
>
> ---
>
> ### **2. Poorly defined figures and metrics**
>
> We respectfully clarify that the concepts of “first-level” and “second-level” alignment are explicitly defined in Section 4.2 (lines 238–240, just two lines about Figure 2):
>
> > *As illustrated in Figure 2, the first level ensures temporal alignment, pairing token sets from the same time step, while the second level ensures structural alignment, matching whole and fractional tokens within each set.*
>
> Equation (4) is complete as written, and the distinction between **whole tokens** and **fractional tokens** is natural and intentional. A numerical value consists of:
> - a **whole number**,
> - a **decimal point**, and
> - a **fractional part**.
>
> After tokenization, these components map into different token groups (whole, decimal, and fractional). Figure 2 visually separates these components, demonstrating clearly why whole tokens cannot be compared with fractional tokens. Therefore, the metric is both coherent and well defined.
>
> ---
>
> ## **3 & 4. The algorithm is heuristic and lacks theoretical guarantees**
>
> We acknowledge that the proposed method is heuristic and empirical, and we cannot provide formal theoretical guarantees. However, this limitation is not unique to our work, which is an inherent challenge shared across:
>
> - NP-hard optimization problems,
> - adversarial attack research, and
> - LLM behavior modeling.
>
> The attack formulation against LLM-based forecasters is **NP-hard**. Unless **P = NP** is proven, no polynomial-time principled solution can be expected. Our dynamic-programming-based method offers a practical heuristic.
>
> Despite its heuristic nature, the attack demonstrates strong empirical performance across ten LLM-based forecasters and six real-world applications, validating the practical effectiveness of our approach.
>
> ---
>
> ### **5. Experimental concerns**
>
> Reviewer GxMJ notes that TDA performs comparably or worse than DGA on some datasets, questioning the consistency of its advantages.
>
> We respectfully clarify a key distinction:
>
> - **DGA is a query-based, black-box attack**, requiring repeated interaction with the forecasting model to iteratively refine perturbations.
> - **TDA is a query-free, black-box attack**, relying solely on the tokenizer, and no model queries are required.
>
> Given this fundamental difference, achieving performance **comparable to a query-based method** is already a strong indication of the strength of TDA. Matching DGA while using **zero model queries** demonstrates that the proposed design is both practical and highly effective in realistic black-box settings.

---

### Official Review · Reviewer_9bBJ · 2025-10-31

**Soundness:** 2
**Presentation:** 3
**Contribution:** 3
**Rating:** 4
**Confidence:** 3

**Summary:**

This paper proposes Token Disruption Attack (TDA), a purely black-box adversarial attack for time-series forecasting. Unlike existing attacks that require repeated interaction with the target system, TDA only needs access to a locally hosted tokenizer, which greatly improves stealthiness. Experiments on multiple LLM-based and non-LLM forecasting models show that TDA induces forecasting errors comparable to those of existing query-based attacks while being more efficient and stealthy.

**Strengths:**

1. Although the proposed technique is not conceptually novel from a general adversarial attack perspective, the authors identify an underexplored attack surface for LLM-based time-series forecasting.
2. TDA operates in a strictly black-box setting and does not require interaction with the target forecasting model.
3. The proposed attack is efficient, showing faster runtime compared with query-based baselines.

**Weaknesses:**

1. In Equation (4), the paper does not clearly define how token differences are computed. If the attack directly compares token ID values, larger ID gaps may not correspond to larger semantic differences, which undermines the validity of the attack's objective.
2. Commercial forecasting services may switch or ensemble LLMs with different tokenizers based on the user's tier and requst property. Since the threat model assumes access to the target tokenizer, it is unclear how TDA performs when the true tokenizer is unknown or mixed. A small experiment on tokenizer mismatch or multi-LLM settings would strengthen the paper.
3. It is unclear whether TDA’s optimization could produce overly long perturbed inputs or concentrate changes on specific numbers in the sequence, which would reduce its stealthiness. The paper does not clearly indicate whether the optimization includes any penalty or regularization term to prevent such behavior.
4. In Algorithm 1, the choice of 30% in lines 8 and 10 is not justified. It appears to be an empirical setting rather than a theoretically grounded choice.
5. Since the optimization objective focuses solely on maximizing token distance differences, there is no theoretical guarantee that this objective would lead to larger forecasting errors. In fact, several cases in Table 1 show that TDA only causes limited error increases.
6. The real-world feasibility of the attack remains unclear. The authors should specify concrete scenarios in which an attacker could realistically modify and optimize forecasting inputs.

**Questions:**

Please check my questions in the Weaknesses section.

---

> ### Author Response · Authors · 2025-12-03
>
> Thanks so much for your time and sincere comments. We address each point in detail below.
>
> ---
>
> ### **1. Token ID gaps vs. semantic differences**
>
> This is an excellent question that highlights a key distinction between LLMs used for natural language understanding and LLM-based time series forecasters.
>
> Our goal is **not** to preserve or manipulate semantic meaning. Time series inputs are structured sequences of numbers, where semantics play a minimal role. Instead, our objective is to **disrupt the tokenization output while maintaining numerical similarity** in the decoded space.
>
> In this context:
>
> - Larger token ID gaps imply larger shifts in the tokenizer’s internal representation.
> - These shifts affect the model’s downstream behavior even though the decoded numeric values remain close.
>
> Thus, token ID gaps are precisely aligned with the goals of a token-level adversarial attack for numerical data.
>
> ---
>
> ### **2. Performance when the true tokenizer is unknown or mixed**
>
> This is an excellent suggestion. In response, we added an empirical study on attack effectiveness under tokenizer mismatch in **Appendix G**. We conduct transfer-based attacks in which perturbations are generated using the tokenizers of ChatGPT 4o and Gemini 2.5 Flash, and then applied to four different LLM-based forecasters. The results show that adversarial perturbations generated from mismatched tokenizers lead to, on average, a 57% smaller error drop compared to the original TDA.
>
> ---
>
> ### **3. Risk of overly long perturbed sequences or concentrated changes**
>
> The length of the perturbed token sequence is inherently bounded by Algorithm 1.
> - The number of iterations equals the length of the clean token sequence.
> - Therefore, the adversarial sequence cannot grow unboundedly.
>
> Our statistics show:
>
> - Over **70%** of poisoned examples have exactly the same length as clean examples.
> - The maximum observed length increase is below **14%**.
>
> Stealthiness is further guaranteed by the $l_2$-norm constraint: during candidate selection, we decode the partial sequence into numeric space and retain only candidates that remain numerically close. Otherwise, the clean token is kept.
>
> ---
>
> ### **4. Empirical choice of 30% in lines 8 and 10 of Algorithm 1**
>
> We agree that this threshold is empirical. Statistical analysis shows that:
>
> - If a numeric token lies on the left side of the vocabulary distribution, the final poisoned token falls within the rightmost 30% with probability **$\approx$97%**, and vice versa.
>
> To avoid unnecessary search and reduce computation, we empirically restrict the candidate region to this 30% band.
>
> ---
>
> ### **5. Lack of theoretical guarantee for increased forecasting error**
>
> This limitation is inherent to the problem setting.
> - Adversarial perturbation generation for black-box LLM-based forecasters is **NP-hard** problem.
> - We further assume that the attacker **cannot query the forecaster**, making theoretical guarantees impossible.
>
> Thus, we provide an empirical, rather than theoretical, solution. Despite its heuristic nature, the proposed TDA demonstrates strong attack performance across 10 LLM-based forecasters and 6 real-world datasets, validating its practical effectiveness.
>
> ---
>
> ### **6. Realistic scenarios where attackers can modify forecasting inputs**
>
> The threat model adopted here follows widely used assumptions in the literature on adversarial attacks for time series forecasting [1–4]. Below are two concrete real-world scenarios:
>
> #### **Traffic speed forecasting (e.g., NeurIPS 2022 [2])**
> Traffic speed is measured by loop detectors or roadside cameras. An attacker controlling a vehicle can subtly adjust driving speed (e.g., ±5 km/h) when passing sensors according to the computed perturbation. Such small adjustments are easy to execute but can significantly distort the sensed traffic state and degrade forecasting accuracy.
>
> #### **Electric load forecasting (e.g., ICFES 2019 [4])**
> Electric load is recorded by household- or building-level meters. A malicious tenant can switch appliances on or off to introduce small load variations (e.g., ±0.1 kW). These minor actions are trivial to perform yet can influence short-term load forecasts.
>
> These examples are widely accepted in prior adversarial time series literature and demonstrate that attackers can realistically manipulate forecasting inputs.
>
> ---
>
> ### **References**
>
> [1] Raphaël, D. et al. *Adversarial Attacks on Probabilistic Autoregressive Forecasting Models*, ICML 2020.
> [2] Liu, F. et al. *Practical Adversarial Attacks on Spatiotemporal Traffic Forecasting Models*, NeurIPS 2022.
> [3] Liu, L. et al. *Robust Multivariate Time-Series Forecasting: Adversarial Attacks and Defense Mechanisms*, ICLR 2023.
> [4] Chen, Y. et al. *Exploiting Vulnerabilities of Load Forecasting through Adversarial Attacks*, ACM ICFES 2019.

---

### Author Response · Authors · 2025-12-03

### **1. Why this submission matters**

**Background**: LLMs have emerged as promising foundation models for forecasting due to their strong zero-shot capabilities. This potential has drawn significant attention from academia, major industry players such as Amazon, and innovative startups like NIXTLA.

**Gap**: Existing adversarial attacks against LLM-based forecasters rely on iteratively querying the target model. This dependence not only introduces computational overhead but also violates stealthiness constraints. Can an attacker manipulate an LLM-based forecaster without querying the model at all?

**Proposed solution**: We propose a novel **query-free** attack mechanism that perturbs LLM-based time series forecasters using only the local tokenizer, with no interaction with the forecasting model during the attack. This design is feasible because most commercial LLMs publicly release their tokenizers, which are easily accessible without permissions or authentication.

**Empirical study**: We conduct a comprehensive evaluation across 10 LLM-based models and 2 non-LLM models spanning 6 real-world forecasting applications. The results show that perturbing as little as 2% of the input data can induce forecasting error increases ranging from 7% to 41%.

**Overall contribution**:
We believe this work is valuable because it uncovers a critical and previously unexploited vulnerability: an attacker can, by accessing only the tokenizer, mount a practical and stealthy poisoning attack that severely undermines the robustness of LLM-based time series forecasters.

---

### **2. Appreciation statement**

We regret that we were unable to engage in real-time discussion with the reviewers, as the discussion period unexpectedly closed due to the ICLR “black swan” event.

Despite the chaos, most of the reviews we received are high-quality and inspiring, which have elevated our submission to the next level. We sincerely thank you for the thoughtful and constructive feedback.

---

### Meta-Review · Area_Chair_ee3M · 2025-12-19

**Summary:**

This paper proposes a black-box adversarial attack named Token Disruption Attack (TDA) for time-series forecasting

Strengths:
(1) New, relatively underexplored attack surface for LLM-based time-series forecasting.
(2) Black-box attack without the interaction with the forecasting models.
(3) Computational efficiency with faster.

Weaknesses:
(1) Lack of theoretic justification for certain choice in the proposed algorithm.
(2) Practical feasibility of the proposed method is unclear.
(3) The threat model is not clearly defined.
(4) Some figures and metrics are not well defined.

**Reviewer Concerns:**

(1) Practical feasibility is partially addressed.
(2) The concerns regarding the figures and metrics are addressed.

**Reviewer Scores:**

While the authors addressed some concerns raised by reviewers, it is unlikely for them to change the scores.

---

### Decision · Program_Chairs · 2026-01-26

Reject